# High Stability and Low Irritation of Retinol Propionate and Hydroxypinacolone Retinoate Supramolecular Nanoparticles with Effective Anti-Wrinkle Efficacy

**DOI:** 10.3390/pharmaceutics15030731

**Published:** 2023-02-22

**Authors:** De Bai, Fan Hu, Huixian Xu, Jiahong Huang, Chengyu Wu, Jiaheng Zhang, Rui Ye

**Affiliations:** 1Sauvage Laboratory for Smart Materials, Harbin Institute of Technology, Shenzhen 518055, China; 2Inertia Shanghai Biotechnology Co., Ltd., Shanghai 200000, China; 3DermaHealth Shanghai Biotechnology Co., Ltd., Shanghai 200000, China; 4Shenzhen Shinesky Biological Technology Co., Ltd., Shenzhen 518055, China

**Keywords:** retinyl propionate, hydroxypinacolone retinoate, gravi-A nanoparticles, high stability, low irritation, transdermal delivery efficiency

## Abstract

Gravi-A nanoparticles, composed of retinyl propionate (RP) and hydroxypinacolone retinoate (HPR), were prepared by encapsulating the two using the high-pressure homogenization technique. The nanoparticles are effective in anti-wrinkle treatment with high stability and low irritation. We evaluated the effect of different process parameters on nanoparticle preparation. Supramolecular technology effectively produced nanoparticles with spherical shapes with an average size of 101.1 nm. The encapsulation efficiency was in the 97.98–98.35% range. The system showed a sustained release profile for reducing the irritation caused by Gravi-A nanoparticles. Furthermore, applying lipid nanoparticle encapsulation technology improved the transdermal efficiency of the nanoparticles, thereby allowing these to penetrate deep into the dermis layer to achieve precise and sustained release of active ingredients. Gravi-A nanoparticles can be extensively and conveniently used in cosmetics and other related formulations by direct application.

## 1. Introduction

With the improvement in living standards and education levels, consumer attention has gradually shifted from the influence of advertisements to the ingredients and preparation process of cosmetics to ensure that beauty products can effectively improve skin conditions [1]. Cosmeceuticals that benefit the human body, especially those used to treat or inhibit skin aging, hold a promising segment in the skincare market [2]. Recently, several drugs with functional ingredients have been mass-produced through biotechnology and drug synthesis methods [3]. Vitamin A (VA) and its derivatives are of interest in the cosmeceutical industry because they act as antioxidants and cell regulators, thereby improving skin texture by stimulating collagen production and reducing skin damage [4].

Retinyl propionate (RP) is a derivative of vitamin A [5]. It is mainly used as a skin conditioner in cosmetics and skin care products and is converted into retinol in the skin [6,7]. Retinol accelerates skin metabolism, promotes cell proliferation, and stimulates collagen production, which is beneficial for acne treatment [8]. Hydroxypinacolone retinoate (HPR) is part of a new generation of anti-aging ingredients in the retinoic acid family, which have unique properties. Unlike other derivatives, HPR can act directly without being converted to retinoic acid, a compound that stimulates the irritation ‘exhibited’ by the skin. It can be safely applied around the eyes with better skin penetration and higher stability [9]. A study by Counts et al [10]. showed that topical application of RP in rats for 14 days resulted in epidermal thickening and enhanced protein and collagen stimulation. In another comparative study, an increase in epidermal thickness was also observed in human skin [11]. Furthermore, our previous study revealed the synergistic anti-aging performance of HPR and RP with a specific weight ratio; thus, we named this combination Gravi-A.

However, HPR and RP are insoluble in water. Because of the existence of all-trans-conjugated double bonds, they are easily degraded and discolored upon exposure to heat and active oxygen. Retinol actives can easily cause local inflammation and skin sensitivity. With side-chain modifications, efficacy is compromised due to the addition of conversion steps [12]. Therefore, improving the stability and efficacy of retinol-based actives and reducing their irritation is essential in cosmetic formulations.

The current strategies for reducing skin irritation caused by retinol active ingredients include encapsulating these in polymeric micelles [13], liposomes [14], and nano-emulsions [15,16] and using slow-release technology. Liposome technology is a superior technology used for cosmetic penetration, wherein a special liposome encapsulates the particles of various active ingredients. In this method, the liposome can efficiently penetrate the basal layer of the skin and be absorbed faster and better than in ordinary methods [17]. When used in basic skin care products, its efficacy is better than in traditional ones. Encapsulated solid, liquid, or gaseous substances in lipid vesicles in the nanosized range can easily penetrate pores and stratum corneum gaps into the deeper layers of the skin, thereby releasing their contents at a controlled rate during application. The skin can absorb more water and nutrients, thus maximizing the replenishment of the lack of dermal nutrient delivery [18]. Nutrients are retained on the stratum corneum in small amounts to protect against the loss of moisture from the skin, thereby making the skin moisturized and full of vitality. Long-term application of skin care products using this technology can delay skin aging. It has been reported that the active components are microencapsulated by various methods. For example, retinoid was successfully encapsulated in albumin using the emulsion method [19], in solid lipid nanoparticles of triglyceride behenate using the melting and curing methods [20], and in silicone particles using microencapsulation for local delivery [21].

In this study, we report the preparation of a new nano-liposome, that is, a liposome-encapsulated VA derivative. The active element is Gravi-A, that is, HPR + RP in a specific weight ratio, which is encapsulated in nanocapsules to suppress the adverse effects of retinol and retinoic acid in cosmetic applications. The size of Gravi-A nanoparticles is approximately 101.1 nm. A nano-emulsion with an average particle size of 101.1 nm can stably suspend the active substance in the nanocarrier and control its release rate while allowing it to effectively penetrate the skin to exhibit its function. Other encapsulation methods, such as coacervation, ionic gels, or inclusion complex formation, require stabilizers/crosslinkers or expensive preparation methods. Thus, encapsulating active components such as retinoids, liposome encapsulation based on supramolecular chemistry can be an economical and environment-friendly method to impart the benefits of retinoids to the skin, thereby improving the transdermal delivery of nanoparticles with minimal use of synthetic chemicals.

## 2. Experimental Section

### 2.1. Materials

Hydroxypinacolone retinoate (HPR, 99%), retinyl propionate (RP, 98%), betaine (98%), glycerol (99%), 1, 2-butanediol (95%), 1, 2-pentanediol (98%), edetate disodium (AR), ethoxylated hydrogenated castor oil, and caprylic/capric triglyceride (≥98%) were purchased from Aladdin and used as received. Glycerol (99%), lecithin from soybean (99%), and vitamin E acetate (96%) were obtained from Macklin Biochemical Co., Ltd., Shanghai, China, and used as received. Poly(lactic-co-glycolic acid) (PLGA, Mn = 20,000) and medium-chain triglycerides were supplied by Yuan ye Bio-Technology Co., Ltd., Shanghai, China. Other common organic solvents were commercially available.

### 2.2. Preparation Methods

#### 2.2.1. Preparation of Gravi-A Nanoparticles

The Gravi-A nanoparticles were prepared according to the conventional emulsion preparation method. Caprylic/capric triglyceride (25.0 wt %), soybean lecithin (3.0 wt %), PEG-40 hydrogenated castor oil (3.0 wt %), and vitamin E acetate (1.0 wt %) were mixed and stirred at 60 °C in an oil bath. HPR (3.0 wt %) and RP (5.4 wt %) were then added to the above solution and stirred and mixed to form the oil phase. Glycerol (5.5 wt %), 2-pentanediol (4.0 wt %), 2-butanediol (5.5 wt %), edetate disodium (1.0 wt %), and purified water (42.5 wt %) were mixed and used as the aqueous phase. Under the condition of shear, the oil phase was slowly added to the aqueous phase and mixed thoroughly. The Gravi-A microcapsules were obtained by adding sodium oleate (1.1 wt %) to the mixture and homogenizing it thrice under high pressure.

#### 2.2.2. Preparation of Free HPR + RP Formulation

The free HPR + RP formulation was prepared as a comparative test as follows: HPR (3.0 wt %), RP (5.4 wt %), caprylic/capric triglyceride (25.0 wt %), soybean lecithin (3.0 wt %), PEG-40 hydrogenated castor oil (3.0 wt %), sodium oleate (1.1 wt %), and vitamin E acetate (1.0 wt %) were stirred and mixed at 60 °C in an oil bath to form the oil phase. Glycerol (5.5 wt %), 2-pentanediol (4.0 wt %), 1, 2-butanediol (5.5 wt %), edetate disodium (1.0 wt %), and purified water (42.5 wt %) were mixed and used as the aqueous phase. Under the condition of shear, the oil phase was slowly added to the aqueous phase and mixed thoroughly. The mixture was stirred to form an emulsion and then cooled to obtain a free HPR + RP formulation.

### 2.3. Characterization of Gravi-A Nanoparticles

The mean particle size and zeta potential were determined by the dynamic light scattering (DLS) technique. A Malvern Zetasizer Nano ZS system (Malvern Instruments Ltd., Malvern, Worcestershire, UK) was used for the DLS technique. Measurements were performed at 25 ± 0.1 °C and a scattering angle of 173° with a laser wavelength of 633 nm. Each sample was filtered through a Millipore Millex-LG syringe filter (Merck Millipore, Shanghai, China) with a pore size of 0.45 μm to remove dust and contaminants. The samples were allowed to remain in equilibrium for 10 min before particle size measurements were performed. The number of nanoparticles permeating through the membrane was analyzed using high-performance liquid chromatography (HPLC, UltiMate 3000, Dionex, Thermofisher, Waltham, MA, USA) with a reversed-phase column (Jupiter 5 μm C18 300 Å, 250 × 4.6 mm, Phenomenex, Los Angeles, CA, USA).

Stability of Gravi-A nanoparticles: The physical stability of Gravi-A nanoparticles at separate times and temperatures was determined by measuring the size of their droplets using DLS. The physical stability was confirmed via centrifugation at 10,000× *g* at 25 °C for 20 min. The drug loading and encapsulation efficiencies of Gravi-A nanoparticles determined the chemical stability of the formulations.

### 2.4. Cytotoxicity Evaluation

Human dermal fibroblasts were provided by Boxi Biotechnology Co., Ltd., Guangdong, China. The toxicity of the free HPR + RP and supramolecular Gravi-A microlipid capsule was evaluated via 3-(4,5-dimethylthiazol-2-yl)-2,5-diphenyltetrazolium bromide (MTT) assay. Briefly, human dermal fibroblasts (1 × 10^4^ CFUs/mL) in 15% fetal calf serum medium were cultured in a 24-well plate for 48 h. The sample solution to be tested was placed into the fibroblast cell solution, 100 μL per well, and cultured at 37 °C for 24 h; 10 μL of MTT solution (5 g/L in phosphate-buffered saline, PBS) was added into each well. After being kept for another 4 h at 37 °C, the MTT medium was removed; 150 μL of dimethyl sulfoxide (DMSO) was added to each well to dissolve any formazan crystals. The cell viability was measured by recording the absorbance of each well at 490 nm (Bio Tek, Winusky, VT, USA). All the measurements were repeated three times. The relative growth rate (RGR) of the human dermal fibroblast cells was calculated according to the following formula with phosphatidylcholine (PC, 10% in DMSO) as control:(1)$$RGB\left(\%\right)=\frac{TestOD_{490\;\mathrm{nm}}}{NegOD_{490\;\mathrm{nm}}}\times 100$$

### 2.5. In Vitro Transdermal Diffusion Test

In vitro transdermal experiments were performed using Franz diffusion cells. The skin of a small fragrant pig, without damage, was placed under a dissecting microscope. Six pieces of identical sizes were cut from the skin and washed once with PBS; the surface moisture was dried with filter paper. The skin was fixed on the Franz diffusion cell, with the horny layer facing the drug delivery chamber and the dermis facing the receiving pool. An amount of 6.5 mL PBS was added to the receiving chamber, and it was ensured that there were no air bubbles between the dermis and the receiving solution. The instrument was switched on beforehand, the water bath temperature was adjusted to 32 ± 1 °C, and the stirring speed was set to 300 rpm; 2 mL of medicine was administered in the administration room, the film was sealed, and tin foil was added to prevent the liquid from evaporating. The effective permeation area was 0.36 π cm^2^. The skin was shredded and extracted at specific times (0.5, 1, 2, and 4 h). After filtration with 0.22 μm organic filter membrane, the skin was analyzed using HPLC, and the intradermal retention was calculated as follows:(2)$$Q_{S}\left({\mu g\;\mathrm{cm}}^{-2}\right)=C_{sn}\times \frac{V_{S}}{A_{S}}+\sum_{i=1}^{n-1}C_{Si}\times \frac{S}{A_{S}}$$
where $Q_{S}$ is the cumulative amount of drug permeation per unit area (μg cm^−2^), $C_{sn}\;$is the measured drug concentration (μg mL^−1^) in the receptor fluid at *n* sampling intervals, $V_{S}$ is the volume of the receptor pool, $\sum_{i=1}^{n-1}C_{Si}$ is the cumulative drug concentration in the receptor fluid, S is the sampling volume, and $A_{S}$ is the effective diffusion area. Finally, the percutaneous skin was rinsed repeatedly with buffer (PBS). Pathological examination was carried out 24 h later, and part of the skin was taken for immunohistochemical staining, frozen sections, and laser confocal photography.

### 2.6. Repeated Skin Irritation Experiments

Four normal New Zealand rabbits, weighing 1.5–2.0 kg, were provided by Huadong Xinhua Experimental Animal Farm in Huadu District, Guangzhou, China. The skin was confirmed to be normal before the experiment. According to the procedures of the skin irritation test (5.4, repeated skin irritation test in chapter VI of Cosmetic Safety Technical Specifications (2015 edition)), the hairs on both sides of the back spine of the rabbits were cut off before the test. The hair removal range was approximately 3 cm × 3 cm for the left and right, respectively. Approximately 0.5 g of the test substance was smeared on the left and the right depilated skin as a blank control. The test substance was applied once a day, and the application was continued for fourteen days. From the second day, the hair was sheared before each application, and the residual subjects were removed using water. The results were observed 1 h later and scored according to the skin irritation response.

### 2.7. Chick Embryo Chorioallantoic Membrane (CAM) Vascular Test

Specific pathogen-free (SPF) white Laihang chicken embryos were purchased from Boehringer Ingelhan Weitong Biotechnology Co., Ltd., Beijing, China. Their surface was cleaned with 75% alcohol and incubated at 37.5 ± 0.5 °C. Light was used to check the position of the embryo. A small hole was made at the small end of the chicken embryo. Using a syringe, 2.5–3.0 mL egg whites were carefully extracted, and the hole was sealed with collodion. A rectangular window was opened above the embryo. The gap was sealed with a transparent film. On the fourteenth day, the chicken embryos were taken out from the incubator, the membrane was removed, and a Teflon ring was placed on the active area for the application of the drug. Three chicken embryos were used for each concentration, and 40 μL of the test samples were added to the ring. The chicken embryos were placed in the incubator for 30 ± 5 min. After incubation, the chicken embryos were removed from the incubator, the changes in the CAM vessels inside and outside the ring were observed and compared, and the injury to the blood vessels inside the ring was evaluated. Sodium dodecyl sulfate (SDS, 0.05 M) was used as the positive control, and ultrapure water or olive oil was used as the negative control group.

### 2.8. Test for Anti-Wrinkle Efficacy

Human skin fibroblasts were purchased from Boxi Biotechnology Co., Ltd., Guangdong, China, and a human type I collagen enzyme-linked immunosorbent assay (ELISA) kit was provided by Huamei Bioengineering Co., Ltd., Wuhan, China. The cells were inoculated into a 96-well plate with a density of 5.0 × 10^3^ cells per well and cultured for 24 h. When the cell fusion rate in the 96-well plate reached 40–60%, the drug was added in groups, and 250 μL of the sample was added to each well and allowed to stand for 72 h. The supernatant of cells was collected, and the content of human type I collagen was determined by the ELISA kit.

### 2.9. Clinical Trial Efficacy Evaluation

#### 2.9.1. Subjects

Thirty-two subjects aged 18–60 years old, who were not sensitive to commonly used daily chemical products and did not participate in other clinical research projects in the last three months, were selected.

#### 2.9.2. Restrictions

During the test period, treatment of the impact test on the tested area was prohibited. The use of other skin care products affecting the test site was prohibited. Products such as bath lotion, soap, shampoo, and other daily necessities were the same as before the beginning of the test, and no change was allowed.

#### 2.9.3. Test Method

Appropriate spot test equipment was selected using the closed patch test method; 0.020–0.025 g of the double-A microcapsule with essence milk was placed in the spot test equipment; the hypersensitive tape was applied to the curved side of the subject’s forearm, and it was removed after 24 h. The skin reaction was observed at 0.5, 24, and 48 h after tape removal, and the results were recorded according to the skin reaction classification standard in the Cosmetic Technical Standard (2015 edition).

### 2.10. Statistical Analysis

The results are presented as the mean standard deviation. One-way analysis of variance was used to determine the significance of differences. Statistical significance was set at * *p* < 0.05 and** *p* < 0.01.

## 3. Results and Discussion

### 3.1. Characterization, and Physicochemical Properties of Gravi-A Nanoparticles

As shown in Figure 1a, the structures enclosed in the pink box are capable of producing similar effects. The changes marked in red make these different in terms of irritation, solubility, and potency strength. After high-pressure homogenization, Gravi-A nanoparticles appeared as yellow opaque liquid with good fluidity and could be mixed with water in any proportion and diluted to transform into pale blue opaline. The flowing yellow transparent liquid without high-pressure homogenization is free HPR + RP. Transmission electron microscopy shows the spherical nature of Gravi-A nanoparticles (Appendix A) with an average particle size of 101 ± 3.7 nm measured by DLS (Figure 1b). These nanoparticles were suitable for transdermal delivery [22,23]. In addition, the potential of Gravi-A nanoparticles was −38.6 mV, which indicates strong electrostatic repulsion and ensures its good stability in an aqueous solution.

The physicochemical properties of Gravi-A nanoparticles, such as the viscosity, density, pH value, electrical conductivity, and appearance, are listed in Appendix A. The microcapsule exhibited a degree of fluidity of 13.2 mPa·s, nearly neutral pH (6.82) without causing skin irritation, a slightly higher density than water (1.015 g/cm^3^), and a conductivity of 329 μs/cm. Thus, Gravi-A nanoparticles exhibited good stability and physicochemical properties, which are suitable for application as a cosmetic precursor. Encapsulation rate is the key quality attribute of liposomes, which reflects the degree of encapsulation of drugs in the liposome, and enables the improvement of preparation technology. The HPLC results (Appendix A) show an encapsulation rate of 97.98% for HPR by Gravi-A nanoparticles. The rate for RP was 98.35%, indicating the high drug-loading capacity of Gravi-A nanoparticles.

### 3.2. Stability of the Gravi-A Nanoparticles

The stability of Gravi-A nanoparticles was evaluated at various temperatures (−20–45 °C) and periods (0–12 weeks). Gravi-A nanoparticles showed no obvious precipitation, sedimentation, turbidity, or other phenomena for the entire stability test. Moreover, DLS results showed that the dispersion index of Gravi-A nanoparticles was good, and the particle size did not significantly change at different temperatures and periods. It indicates that the as-prepared Gravi-A nanoparticles exhibit good stability (Figure 2a). Notably, the tendency of particle aggregation increased with increasing temperature. Furthermore, we tested the stability of the content of active components in the lipid microcapsule and found that the amount of HPR and RP did not change significantly in the wide temperature range of −20–35 °C, and the content of active components was more than 90% after 12 weeks (Figure 2b,c). At a higher temperature of 45 °C, the value can be stable. However, for uncoated HPR and RP, their stability at 45 °C significantly decreased. After 12 weeks, 74% and 62% of HPR and RP remained, respectively. Some useful information can be obtained from the comparison diagram. Compared with free HPR and RP, the stability of the content of active components in nanoparticles is enhanced at the same temperature. The results showed that the Gravi-A nanoparticles were highly stable and suitable for most normal environments. Notably, neither Gravi-A nanoparticles nor free Gravi-A showed a detectable release of retinoic acid during the stability test, indicating the superior stability and safety of the HPR and RP combination.

### 3.3. Cytotoxicity of the Free HPR + RP and Gravi-A Nanoparticles

Low toxicity of cosmetics to human cells is necessary. As shown in Figure 3, the cytotoxicity of the free HPR + RP and supramolecular Gravi-A nanoparticles was evaluated by MTT assay. All subjects were incubated with fibroblast cells. Based on the absorbance values at 490 nm from the MTT assay, the relative growth rates (RGRs) of the dermal fibroblast cells were calculated. The trend of relative cell survival rate is shown in Figure 3. Notably, the relative toxicity of free HPR + RP is higher than that of encapsulated Gravi-A nanoparticles. For Gravi-A nanoparticles, the toxic effect was observed in a dose- and time-dependent manner. With the increase in sample volume concentration, the relative survival rate of cells decreased gradually. The highest relative survival rate was 114.69% at 0.0078% (*v*/*v*) (Figure 3a). The same trend is observed for free HPR + RP (Figure 3b). Based on human skin fibroblasts, the IC_50_ of Gravi-A nanoparticles was 8.933%, and that of free HPR + RP was 1.308% (Figure 3c). The results showed that both Gravi-A nanoparticles and free HPR + RP emulsions were almost non-toxic to human fibroblasts.

### 3.4. Cellular Uptake and Skin Penetration of the Free HPR + RP and Gravi-A Nanoparticles

A porcine (Sus scrofa f. domestica) skin model was used to measure the amount of retention of Gravi-A on the skin at separate times. Hematoxylin-eosin (H&E) staining, frozen section, and CLSM were performed to characterize the transdermal delivery efficiency of the major components in the sample. Because stratum corneum is a major barrier to drug transport, in vitro permeation experiments using Franz diffusion cells and an ex vivo model are simple and effective methods for evaluating transdermal delivery [24]. The transdermal absorption of HPR and RP was evaluated by CLSM analysis. The fluorescence intensity of free HPR + RP was weak and could not be detected in the deep layer of the skin. In contrast, the fluorescence intensity of the encapsulated Gravi-A significantly increased and was distributed throughout the skin layer (Figure 4a). Figure 4b shows the corresponding fluorescence intensity of the sample. The transdermal penetration of Gravi-A nanoparticles was 3.57 times higher than that of free HPR + RP. In vitro permeability statistics (Appendix A) show that the amount of cumulative release of the free group reaches 100% within 4 h, and the release rate slowed down after packaging, thereby achieving the accurate and continuous release of packaged components after shell degradation.

Skin irritation is a common side effect of transdermal delivery; therefore, H&E staining was performed on the skin after an in vitro permeation experiment, and no skin damage or irritation was observed (Figure 4c). Figure 4d shows the intradermal retention of actives at different times. After the skin was treated with samples for 0.5, 1, 2, and 4 h, compared with the active HPR in the encapsulated and non-encapsulated samples, the intradermal retention per unit area of the wrapped HPR sample was approximately 52.90 μg/cm^2,^ and that of the free HPR sample was approximately 12.30 μg/cm^2^ after 4h incubation. Thus, the transdermal efficiency of encapsulated HPR is four times that of free HPR. Similarly, the transdermal efficiency of encapsulated RP is five times that of free RP. The result indicates that the encapsulation carrier and supramolecular solvent can improve the transdermal efficiency of nanoparticles and assist HPR and RP penetrate the cuticle layer of the skin.

### 3.5. Multiple Skin Irritation Tests

Multiple skin irritation tests exhibit the effect of chronic cumulative irritants of the subject on the skin, which is an important toxicological index for evaluating the hygienic quality of cosmetics. After 14 days of continuous application of Gravi-A nanoparticles, the average score of each animal per day was calculated according to the observation score of skin irritation reaction in Appendix A, and no irritation reaction was observed in the blank control area. The observations of the test area are shown in Appendix A. The average score of each animal per day was 0.68. Based on the classification of skin irritation intensity, the intensity of multiple skin irritations caused by double-A nanoparticles were classified as mild irritation. For free HPR + RP, the average score per animal per day was 0.84 (Appendix A), which also showed slight irritation. These results indicate the safety basis for the clinical application of the product.

### 3.6. Vascular Experiment of Chick Embryo CAM

The chorioallantoic membrane (CAM) is a respiratory membrane that surrounds chicken embryos. In this experiment, taking advantage of the integrity, clarity and transparency of the vascular system of the chorioallantoic membrane in the middle stage of the hatched chicken embryo, a certain amount of samples were directly contacted with the chicken embryo allantoic membrane, and the changes of the toxic effect index of the chorioallantoic membrane were observed after a period of time. Used to evaluate the eye irritation of the sample.

As shown in Figure 5a, when the sample concentration was 50% and 80%, the blood vessels in the chorioallantoic membrane of the chicken embryo could be observed clearly and normally before sample addition, and there was no significant change after 30min. When the sample concentration was 100%, the blood vessels showed slight bleeding. The value of RC_50_ calculated by probability analysis is 94.51%. Similarly, the same conclusion can be obtained in the free HPR + RP experiment (Figure 5b,c). It has no irritation to human eyes and can be used as an active ingredient in human skin.

### 3.7. Determination of Type I Collagen Amount in Human Skin Fibroblasts

Based on human skin fibroblasts, the firming efficiency of the sample was evaluated by detecting changes in the amount of human type I collagen synthesis. Type I collagen is the main component of the extracellular matrix of the dermis, which is synthesized intracellularly by dermal fibroblasts and secreted extracellularly by type I collagen. Under the action of terminal procollagen peptidase, the terminal peptides are separated and polymerized to form collagen fibers. Therefore, the firmness and anti-wrinkle ability of the sample can be evaluated by the amount of type I collagen secreted by fibroblasts after administration. The T-test method was used for analysis, as shown in Figure 6a. Compared with the negative control (NC) group, the upregulation rate of human type I collagen amount was 226% (*p* < 0.05), 194% (*p* < 0.05), and 217% (*p* < 0.05) at the test concentrations of 0.01%, 0.008%, and 0.006% of double-A nanoparticles, respectively. Thus, Gravi-A nanoparticles can promote the synthesis of human type I collagen in this concentration range, and the ideal anti-wrinkle and firming effect can be achieved. According to the experimental results shown in Figure 6b, it is confirmed that free HPR + RP also can promote the synthesis of human type I collagen in a low concentration range; however, the upregulation rate of human type I collagen amount was lower than that in Gravi-A nanoparticles. The results show that human skin easily absorbs the coated Gravi-A nanoparticles, further improving its anti-wrinkle effect.

### 3.8. Clinical Safety and Efficacy Evaluation

Based on the above results, a clinical efficacy evaluation was conducted [25,26,27]. All tests were performed following ethical regulations. We used professional skin testing equipment VISIA to test several volunteers of different ages after using Gravi-A nanoparticles to track the nasolabial folds, crow’s feet, forehead wrinkles, cervical stripe, and frown lines. After two months of testing, the nasolabial folds, crow’s feet, and frown lines decreased by 27.57%, 31.15%, and 21.17%, respectively. The effect on the elimination of forehead wrinkles was the most significant, and the area decreased by 56.28% after 56 days (Figure 7a). Correspondingly, the depth of nasolabial folds and forehead wrinkles, the length of crow’s feet and frown lines, and the width of the cervical stripe were reduced, and the length of frown lines was reduced by 76.46%. Furthermore, the carrier for transdermal drug delivery could not damage the skin barrier while delivering the drug efficiently. For a specific comparison of before and after wrinkle removal, see the supporting material (Appendix A). As shown in Figure 7b, after 28 days of clinical application, Gravi-A nanoparticles significantly reduced the eye wrinkles of volunteers. In addition, it was found that the depth of skin wrinkles decreased by 42.58% and the area of wrinkles decreased by 32.51% (Figure 7c). Not only is Gravi-A nanoparticle effective in reducing various wrinkles, but it also improves skin firmness and elasticity. For example, skin firmness (F4) started showing a significant reduction from Week 2 of using the product (Appendix A). F4 value decreased by 17.23% at Week 2 and continued to decrease by 35.18% at Week 8 compared to baseline. Furthermore, skin elasticity (R2) started showing significant improvement from Week 1 (Appendix A). R2 value increased by 5.19% at Week 1 compared to baseline, whereas, after 28 and 56 days of product use, R2 value increased by 21.84% and 17.64%, respectively.

These results were reproducible in multiple experimental groups, indicating that the synthesized Gravi-A nanoparticles increased skin firmness and elasticity and inhibited wrinkle formation without adverse effects, such as irritation or toxicity to the skin. In summary, the synthesized Gravi-A nanoparticles show high clinical safety and outstanding clinical efficacy for antioxidant, anti-wrinkle, moisturizing, and repairing properties.

## 4. Conclusions

HPR exhibits good gene expression and stability, which can play an immediate anti-aging effect, while RP exhibits a good balance to achieve safety and efficacy. The supramolecular Gravi-A nanoparticles prepared using HPR and RP can be mixed with water in any proportion, which can not only improve the stability but also reduce the irritation of HPR and RP alone and improve the transdermal delivery efficiency. In conclusion, the combined encapsulated HPR and RP (Gravi-A) nanoparticles showed good stability and high drug-loading capacity. Compared to the free Gravi-A, the encapsulated Gravi-A showed lower IC_50_ in the fibroblast model and higher skin retention and deeper penetration in the porcine skin model, indicating higher safety and better anti-aging effects after encapsulation. The application of this encapsulation in fibroblast-induced collagen production and clinical study further demonstrated its remarkable anti-aging efficacy with the reduction of wrinkles and increase in elasticity.

## Figures and Tables

**Figure 1 pharmaceutics-15-00731-f001:**
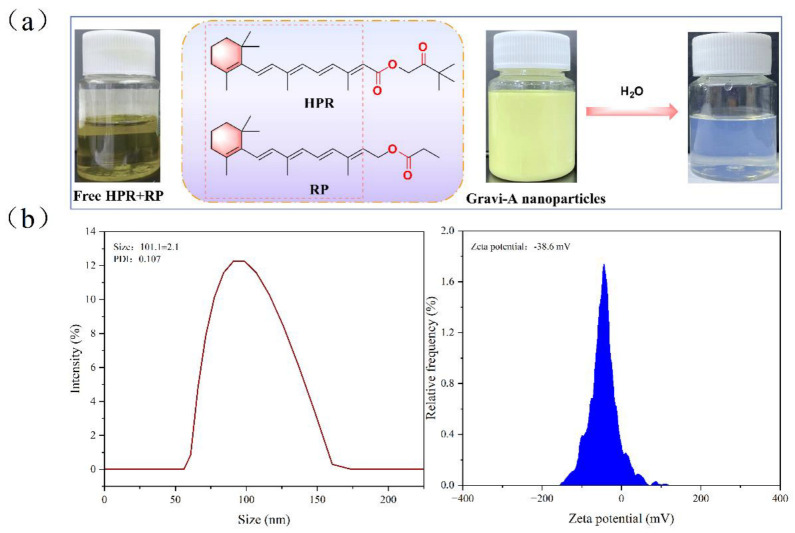
(**a**) Free HPR + RP and supramolecular Gravi−A nanoparticles, where PDI represents polymer dispersion index. (**b**) Particle size and zeta potential of supramolecular Gravi−A nanoparticles using DLS.

**Figure 2 pharmaceutics-15-00731-f002:**
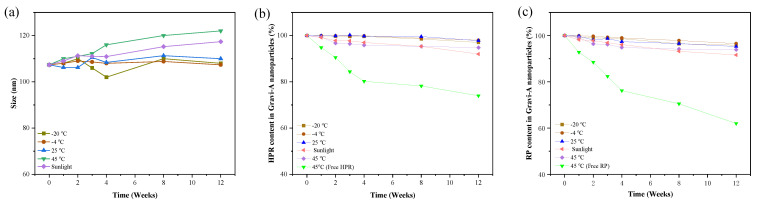
(**a**) Particle size stability. Content stability of (**b**) HPR and (**c**) RP in Gravi−A nanoparticles subjected to different temperatures or periods.

**Figure 3 pharmaceutics-15-00731-f003:**
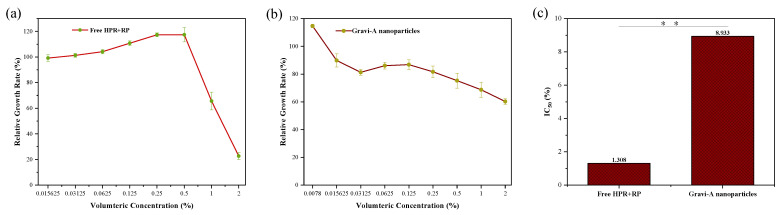
Relative growth rates (RGR) of (**a**) free HPR + RP and (**b**) supramolecular Gravi-A nanoparticles. (**c**) Human dermal fibroblast cells detected by MTT assay (** *p* < 0.01).

**Figure 4 pharmaceutics-15-00731-f004:**
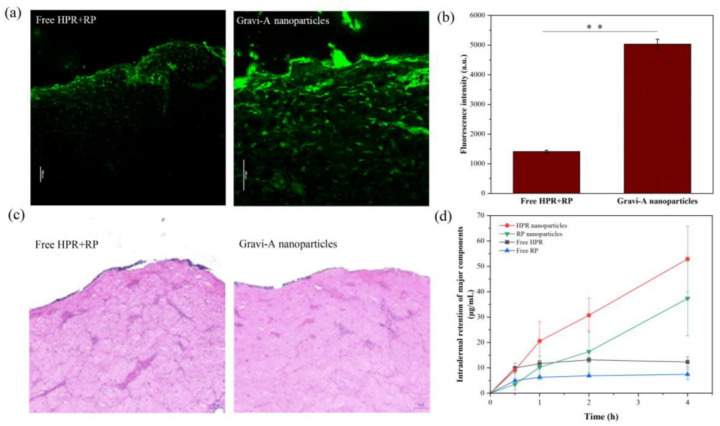
Skin penetration and biocompatibility of free HPR + RP and supramolecular Gravi−A nanoparticles. (**a**) CLSM images (scale bars: 100 μm); (**b**) Comparison of fluorescence intensity; (**c**) H&E-stained sections of porcine skin after treatment (scale bars: 100 μm, ** *p* < 0.01); (**d**) Cumulative penetration of the main components in porcine skin.

**Figure 5 pharmaceutics-15-00731-f005:**
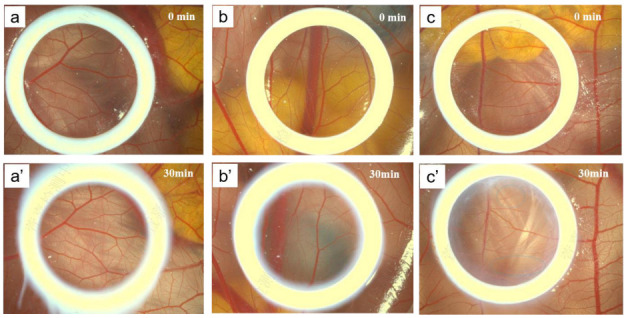
Effect of supramolecular Gravi-A nanoparticles on chorioallantoic membrane toxicity of chicken embryo after 30 min administration at different concentrations, where the yellow circle represents the selected viewing area. A total of 2.5% sample concentration in group (**a**,**a’**), 4% sample concentration in group (**b**,**b’**)**,** and 5% sample concentration in group (**c**,**c’**).

**Figure 6 pharmaceutics-15-00731-f006:**
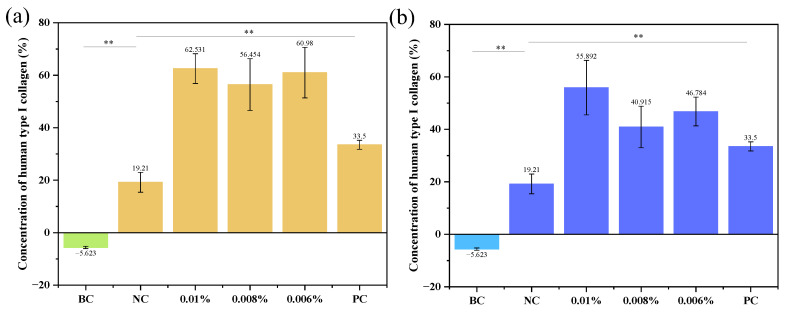
Human type I collagen content. (**a**) The detection results of human type I collagen content based on human skin fibroblast model under different concentrations of Gravi-A nanoparticles. (**b**) The detection results of human type I collagen content based on human skin fibroblast model with free HPR + RP at different concentrations. Where BC stands for blank control; NC represents negative control, complete culture medium incubation; PC represents a positive control and was incubated with a medium containing 100 ng/mL TGF-β1; ** *p* < 0.01.

**Figure 7 pharmaceutics-15-00731-f007:**
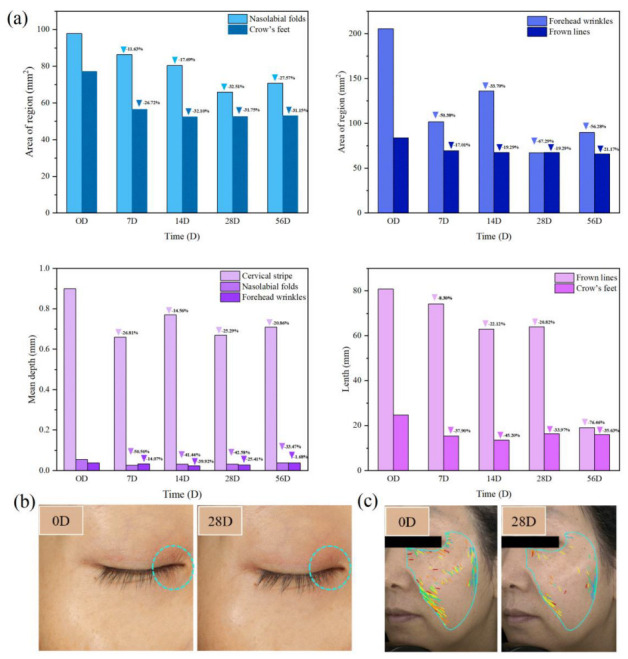
(**a**) Year-on-year changes in facial wrinkles after using Gravi−A nanoparticles for 56 days, where triangles/arrow heads represent the wrinkle area is gradually decreasing compared with 0 D. VISIA shows the anti-wrinkle effect of the control group and Gravi−A nanoparticles in humans after 28 days of treatment for (**b**) eye wrinkles and (**c**) facial wrinkles, respectively. The blue circle in Figure (**b**) indicates the obvious change area, and in Figure (**c**), light blue lines represent freckles, green lines represent wrinkles, red lines represent inflammation, and yellow lines represent porphyrins.

## Data Availability

All the relevant data are available from the corresponding authors upon reasonable request.

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
