# Peer review of "High Stability and Low Irritation of Retinol Propionate and Hydroxypinacolone Retinoate Supramolecular Nanoparticles with Effective Anti-Wrinkle Efficacy"

_pharmaceutics, 2023, doi:10.3390/pharmaceutics15030731_

Round 1

Reviewer 1 Report

The paper is in my opinion eligible to be published, as the topic is very interesting and the results well argumented.

Author Response

Dear reviewer,

Thank you for your comment. We were pleased to know that our work was rated as potentially acceptable for publication in Journal. We thank you for your time and effort in reviewing the previous version of the manuscript. We would like also to thank you for allowing us to resubmit a revised copy of the manuscript.

We hope that the revised manuscript is accepted for publication in the Journal of Pharmaceutics.

Sincerely,

Zhang

Reviewer 2 Report

The paper "High stability and low irritation of Retinol Propionate and Hy-2 droxypinacolone Retinoate Supramolecular Nanoparticles with 3 effective Anti-wrinkle Efficacy" presents the obtaining and results of the physical-chemical determinations, respectively the effectiveness tests on the skin. The work represents a complex study with clear results that demonstrate the applicative efficiency of new synthesized variant. The work presents Cytotoxicity Evaluation, transdermal diffusion test, determination of skin irritation, anti-wrinkle efficacy, and Clinical trial efficacy evaluation. The results are clearly presented and the applied methodology is well argued.

Author Response

Dear reviewer,

Thank you for your comment. We were pleased to know that our work was rated as potentially acceptable for publication in Journal, subject to minor revision. We thank you for your time and effort in reviewing the previous version of the manuscript. After reading your suggestion, we carefully reviewed the manuscript and deleted the reference 14 which is not much related to the article. Your suggestions have enabled us to improve our work.  We would like also to thank you for allowing us to resubmit a revised copy of the manuscript.

We hope that the revised manuscript is accepted for publication in the Journal of Pharmaceutics.

Sincerely,

Zhang

Reviewer 3 Report

This paper explores a method for producing nanoparticles containing Retinyl propionate (RP) and Hydroxypinacolone retinoate (HPR) for antiwrinkle treatment whilst avoiding the skin irritation sometimes associated with these ingredients. The active ingredients are encapsulated in liposomes.

The preparation methods, physicochemical analyses, and laboratory and clinical trials are described in appropriate detail. The resistance of the nanoparticles to environmental degradation is clearly shown, as is the improved transdermal transmission.

The use of a logarithmic scale for particle size in Figure 1(b) makes it harder for the reader to interpret than would a linear scale.

  This is a well-written and informative paper which merits publication.

Reviewer 4 Report

The authors have reported the creation of a new formulation with a nanoparticle shape composed of retinyl propionate and hydroxypinacolone retinoate for anti-wrinkle remedy with mild, effective, and stable drug release. These results will be helpful and informative for researchers in the field of biomaterials. Furthermore, the reviewer thinks that the authors’ study in this manuscript is quite interesting, suggestive, and well-organized. Therefore, the authors’ manuscript is suitable for publication in “Pharmaceutics” in the present form.

Author Response

(The authors gave the same response as above.)
